# Human Alcohol-Microbiota Mice have Increased Susceptibility to Bacterial Pneumonia

**DOI:** 10.3390/cells12182267

**Published:** 2023-09-13

**Authors:** Kelly C. Cunningham, Deandra R. Smith, Daniel N. Villageliú, Christi M. Ellis, Amanda E. Ramer-Tait, Jeffrey D. Price, Todd A. Wyatt, Daren L. Knoell, Mystera M. Samuelson, Patricia E. Molina, David A. Welsh, Derrick R. Samuelson

**Affiliations:** 1Department of Internal Medicine-Pulmonary Division, College of Medicine, University of Nebraska Medical Center, Omaha, NE 68198, USA; 2Department of Pharmacy Practice and Science, College of Pharmacy, University of Nebraska Medical Center, Omaha, NE 68198, USA; 3Department of Food Science and Technology, University of Nebraska-Lincoln, Lincoln, NE 68588, USA; 4Nebraska Food for Health Center, University of Nebraska-Lincoln, Lincoln, NE 68588, USA; 5Department of Environmental, Agricultural and Occupational Health, College of Public Health, University of Nebraska Medical Center, Omaha, NE 68198, USA; 6Department of Veterans Affairs Nebraska-Western Iowa Health Care System, Omaha, NE 68198, USA; 7Animal Behavior Core, University of Nebraska Medical Center, Omaha, NE 68198, USA; 8Department of Physiology, Louisiana State University Health Sciences Center, New Orleans, LA 70112, USA; 9Department of Internal Medicine, Section of Pulmonary/Critical Care & Allergy/Immunology, Louisiana State University Health Sciences Center, New Orleans, LA 70112, USA

**Keywords:** alcohol, microbiome, gut–lung axis, pneumonia, host defense, streptococcus, klebsiella

## Abstract

Preclinical studies have shown that chronic alcohol abuse leads to alterations in the gastrointestinal microbiota that are associated with behavior changes, physiological alterations, and immunological effects. However, such studies have been limited in their ability to evaluate the direct effects of alcohol-associated dysbiosis. To address this, we developed a humanized alcohol-microbiota mouse model to systematically evaluate the immunological effects of chronic alcohol abuse mediated by intestinal dysbiosis. Germ-free mice were colonized with human fecal microbiota from individuals with high and low Alcohol Use Disorders Identification Test (AUDIT) scores and bred to produce human alcohol-associated microbiota or human control-microbiota F1 progenies. F1 offspring colonized with fecal microbiota from individuals with high AUDIT scores had increased susceptibility to *Klebsiella pneumoniae* and *Streptococcus pneumoniae* pneumonia, as determined by increased mortality rates, pulmonary bacterial burden, and post-infection lung damage. These findings highlight the importance of considering both the direct effects of alcohol and alcohol-induced dysbiosis when investigating the mechanisms behind alcohol-related disorders and treatment strategies.

## 1. Introduction

Numerous preclinical studies have demonstrated that the intestinal microbiota plays an important role in regulating the behavioral outcomes and tissue injury associated with chronic alcohol use. Changes in the gastrointestinal microbiota metabolic function and composition due to alcohol consumption are associated with (1) behavior consequences, (2) physiological alterations, and (3) immunological effects [1,2,3,4,5,6,7]. However, these studies are limited in their ability to assess the direct effects of alcohol-associated dysbiosis independent of the direct effects of alcohol, as they rely either on association studies or on manipulating the microbiota with antibiotics, which is known to have off-target effects [8]. As such, there are currently no animal models that allow for the systematic evaluation of the effects of chronic alcohol abuse that are mediated by alcohol-associated dysbiosis independent of the direct effects of alcohol. This is especially true for evaluating the effects of human alcohol-associated dysbiosis, as only a handful of studies have sought to understand the functional consequences of human alcohol-associated dysbiosis [1,5,9,10]. To address this challenge and knowledge gap, we developed a translationally relevant human alcohol-microbiota mouse model, which allows us to evaluate the immunological effects of alcohol-associated dysbiosis independent of the direct effects of alcohol. Precisely, C57BL/6 germ-free mice were colonized with human fecal microbiota from individuals with high and low Alcohol Use Disorders Identification Test (AUDIT) scores and bred to produce human alcohol-associated microbiota or human control-microbiota F1 progenies. Both female and male progeny (F1 mice), designated as human alcohol-microbiota mice or human control-microbiota mice, were then generated. Evaluation of the intestinal microbial community structure of germ-free mice colonized with human microbiota found that the recipient mice cluster with respect to the original human donor sample. In addition, the F1 generation of human alcohol-microbiota mice or human control-microbiota mice maintained a similar microbial community structure. Finally, utilizing human alcohol-microbiota-associated F1 mice, we found that mice colonized with the fecal microbiota from AUDIT > 8 individuals had increased susceptibility to both *Klebsiella pneumoniae* and *Streptococcus pneumoniae* when compared to mice recolonized with fecal microbiota from AUDIT < 8 individuals, as determined by increased (A) mortality, (B) pulmonary bacterial burden, and (C) lung damage/leak post infection. These data support the use of human samples as well as F1 mice to evaluate the effects of the alcohol-associated dysbiosis on pulmonary host defense, independent of alcohol’s effects on tissues. Further, these findings highlight the importance of considering the direct effects of both alcohol and alcohol-induced dysbiosis when investigating the mechanisms behind alcohol-related disorders.

## 2. Materials and Methods

### 2.1. Mouse Studies

Mice were housed in an SPF environment under standard social housing conditions in Comparative Medicine at UNMC. Food and water were provided ad libitum. All protocols used in these studies were approved by the Institutional Animal Care and Use Committee of UNMC (IACUC# 20-085-09-FC & 20-084-10-FC). This research protocol is in accordance with the NIH and Office of Laboratory Animal Welfare (OLAW) guidelines.

### 2.2. Human Stool Sample Collection and Fecal Engraftment

All human fecal samples were obtained through the New Orleans Alcohol HIV study (NOAH), an NIH P60-funded center. Selection criteria for the human fecal samples were as follows: AUD positive samples were defined as subjects with an Alcohol Use Disorders Identification Test (AUDIT) score of ≥ 8 for men and ≥ 5 for women, with the last alcohol-containing beverage consumed within the 7 days prior to enrollment. Human fecal samples were collected using the Fecal Aliquot Straw Technique, as described previously [11]. Four or more straws were collected for each fecal sample and stored at −80 °C until shipped or used. All human fecal microbiota samples were treated and prepared under anaerobic conditions. Specifically, frozen fecal samples were homogenized in sterile 10% glycerol phosphate buffered saline (PBS; ThermoFisher Scientific, Cincinnati, OH, USA) and filtered to remove large organic particulate matter. The germ-free mice used in this study were obtained from the Nebraska Gnotobiotic Mouse Program (Lincoln, NE, USA). Engraftment of human fecal samples was performed as described previously [12]. Male and female C57BL/6 germ-free mice were engrafted with fecal microbiota from individuals with high and low AUDIT scores. Specifically, 9 breeding pairs derived from 3 human fecal samples from individuals with an AUDIT score > 8, and 9 breeding pairs derived from 3 human fecal samples from individuals with an AUDIT score < 8, were used to establish our alcohol-microbiota and control-microbiota humanized mice. Human-microbiota mice were maintained in sterile individually ventilated cages for the duration of the study. Human alcohol-microbiota mice and human control-microbiota mice were then bred to produce human-microbiota-associated F1 mice. F1 generation of human alcohol-microbiota mice and human control-microbiota colonized mice were used for all experimental endpoints in this study. F1 offspring at 8–10 weeks were infected with either *K. pneumoniae* or *S. pneumoniae* via oropharyngeal aspiration and sacrificed 48 h post infection.

### 2.3. DNA Sequencing of the 16s rRNA Gene

Sequencing of the 16s rRNA bacterial gene was performed in the Genomics Core at UNMC, as previously described [13].

### 2.4. Sequence Analysis

R and the following R packages were used to process all raw sequencing data: DADA2 v1.1.5, Phyloseq v1.16.2, DESeq2 v1.20.0, microViz v0.10.7, microbiome v1.16.0, microbiomeutilities v1.00.16, and vegan v2.3-5 [14,15,16,17,18,19,20,21]. DADA2 was used to truncate, denoise, chimera-filter, and cluster the sequences into amplicon sequence variants (ASVs). Taxonomic classification of ASVs was performed using the SILVA reference database v132. The estimate_richness function in Phyloseq was used to calculate alpha diversity. Phyloseq and vegan were used to calculate beta-diversity using a distance-based redundancy analysis (dbRDA) on sample-wise Bray–Curtis dissimilarity distances. DESeq2 was used to determine the differentially abundant ASVs.

### 2.5. K. pneumoniae and S. pneumoniae Culture and Infection

*K. pneumoniae* strain 43816, serotype 2 (American Type Culture Collection, Manassas, VA) was grown and prepared as previously described [22]. *S. pneumoniae* strain JWV500 (D39hlpA-gfp-Cam’) was grown and prepared for inoculation as described previously [12]. Oropharyngeal aspiration of mice was performed as described elsewhere [22]. Mice were infected with 1 × 10^3^ colony-forming units (CFU) of *K. pneumoniae* or 4 × 10^8^ CFU of *S. pneumoniae* in 100 µL of PBS. The *K. pneumoniae* and *S. pneumoniae* dose was confirmed by serial dilutions. All mice were sacrificed 48 h post infection.

### 2.6. K. pneumoniae and S. pneumoniae Lung and Spleen Quantification

For quantification of pulmonary and splenic *K. pneumoniae* burden, tissues were homogenized, and serial dilutions of the tissue homogenates were plated on HiCrome Klebsiella Selective Agar plates (Thomas Scientific, Swedesboro, NJ, USA) for incubation at 37 °C for 24 h. The CFU/lung or CFU/spleen was calculated based on standard colony counts. To measure the pulmonary and splenic burden of *S. pneumoniae* in the mice, the *lytA* gene was quantified using real-time quantitative PCR. Primer sequences, probe, and thermocycler run parameters are described elsewhere [13].

### 2.7. Bronchoalveolar Lavage (BAL) Fluid Analyses

BAL fluid was collected from the lungs by lavage with PBS. A Bio-Rad TC20 automated cell counter (Bio-Rad, Hercules, CA, USA) was used to determine the total number of BAL cells. Hema-3 cell staining (ThermoFisher Scientific, Cincinnati, OH) and manual counting was used for BAL cell differential counts. BAL levels of total protein, chemokine, and cytokine were determined using commercially available ELISA kits according to their manufacturers’ instructions (BioLegend, San Diego, CA, USA, R&D Systems, Minneapolis, MN, USA and ThermoFisher Scientific, Cincinnati, OH).

### 2.8. Lung Histology

Whole lungs were inflated with 10% formalin (ThermoFisher Scientific, Cincinnati, OH) to preserve pulmonary architecture. The UNMC Tissue Sciences Core Facility then processed all lungs for histological evaluation, as described previously [12]. Lung inflammation was determined by a blinded pathologist using a previously validated scoring system [23,24].

### 2.9. Behavioral Assessments

Anxiety-like behavior was assessed using a marble-burying approach, as described elsewhere [25]. Precisely, individual mice were placed in a standard cage with 10 marbles evenly spaced throughout the cage. After 30 minutes, the number of marbles buried was measured (a marble was considered buried if 2/3 of the marble was covered with bedding). Drinking preference was assessed using a standard 2-bottle alcohol-preference test [26]. Briefly, mice were presented with two graduated water tubes in their home-cage. One tube contained water while the other contained 10% ethanol. The positions of the tubes were switched at 3-day intervals for a total of 15 days. Drinking preferences were assessed every 24 h and alcohol preference was calculated using the following formula: Drinking preference = ethanol intake (mL)/intake of ethanol + intake of water (total fluid-intake in mL) × 100. Drinking preference was then calculated for each 24 h period and averaged and scored as follows: A score of 50% indicates equality of preference, while ≤ 49% indicates aversion, and, finally, a score of ≥ 51% indicates preference.

### 2.10. Statistics

The R packages vegan and GraphPad Prism version 9.1 (GraphPad Software, La Jolla, CA, USA) were used for statistical analysis. Data were presented as the mean ± standard error of the mean. Results were considered statistically significant if *p* < 0.05 or if the false discovery rate (FDR) q-value < 0.05. The number of replicates and sample size are indicated in each figure. One-way analysis of variance (ANOVA) was used for comparisons between three or more groups. Sidak’s correction for multiple comparisons was applied to group comparisons following ANOVA. A Welch’s T-test was used to evaluate statistically significant differences between two groups. Mandel–Cox modeling was used to analyze survival curves. Microbiome statistics were assessed as follows: Alpha-diversity significance was determined using the aov function in stats. Permutational multivariate analysis of variance (PERMANOVA) via the adonis2 function in vegan was used to determine beta-diversity significance. FDR corrections were applied to group comparisons following PERMANOVA. The DESeq2 R package was used to determine differentially abundant ASVs.

## 3. Results

### 3.1. Microbial Community Structure Is Maintained in the F1 Generation of Human Alcohol-Associated Microbiota Mice

As stated in the methods section, a total of six human donor samples were used to generate F1 human alcohol-microbiota mice or human control-microbiota mice. Specific details of each donor are shown below in Table 1. Our breeding/experimental strategy is shown in Figure 1.

Microbiota composition was assessed following human engraftment into germ-free mice, as well as in the F1 generation of human alcohol-associated microbiota and human control-microbiota mice. β-diversity analysis demonstrated that the microbiota composition differed significantly between the 6 groups of conventionalized mice (F = 8.0288, *p* = 0.00001), as well as between control and alcohol-associated microbiota conventionalized mice (F = 3.3482, *p* = 0.00137). Conversely, analysis of the β-diversity between the human donor sample and the mouse cecal samples (F0 and F1 combined) showed no significant difference for D1023 (F = 2.4793, *p* = 0.09087); D1003 (F = 3.2624, *p* = 0.08281), D1022 (F = 3.5607, *p* = 0.0774), D1935 (F = 1.8977, *p* = 0.08957), D1047 (F = 4.574, *p* = 0.07595), and D1038 (F = 3.6382, *p* = 0.08359) (Figure 2A). However, differences in the relative abundance of microbial taxa were observed (Figure 2B). Importantly, the significant differences in differentially abundant microbial taxa were maintained in F0 and F1 conventionalized mice (Figure 3A–C).

### 3.2. Human Alcohol-Associated Microbiota Mice have Increased Weight Loss and Decreased Survival following K. pneumoniae Infection

To evaluate the effects of human alcohol-associated dysbiosis on survival after *K. pneumoniae* infection, both male and female F1 mice were administered 1 × 10^3^ CFU of *K. pneumoniae*, and survival was evaluated 48 h. post infection. All mice exhibited post-infection weight loss (Figure 4). Human alcohol-associated microbiota mice had a trend toward increased weight loss in the combined group (Figure 4A) and there was no difference between control and alcohol-associated male mice (Figure 4B). However, significant weight loss in F1 human alcohol-associated microbiota female mice (Figure 4C) was observed, suggesting a potential gender effect following *K. pneumoniae* infection. In addition, there were distinct effects on post-infection weight loss with *K. pneumoniae* based on the original human donor (Figure 4D). Both F1 male and female alcohol-associated microbiota mice had decreased survival in comparison to their control-microbiota counterparts (Figure 5). Human alcohol-associated microbiota mice had a significant increase in mortality in the combined group (Figure 5A). However, a nonsignificant decrease in mortality was seen between control and alcohol-associated male mice (Figure 5B). Conversely, a significant increase in mortality in F1 human alcohol-associated microbiota female mice (Figure 5C) was observed, again suggesting a potential gender effect following *K. pneumoniae* infection. Further, mortality post infection with *K. pneumoniae* was associated with the original human donor (Figure 5D). Consistent with increased weight loss, we also observed changes in the physical appearance and activity levels (data not shown) with F1 human alcohol-associated microbiota, particularly in female mice, which had the most significant symptoms.

### 3.3. Human Alcohol-Associated Microbiota Mice have Altered Lung Tissue Integrity and Increased Bacterial Burden

To determine the role of human alcohol-associated dysbiosis on pulmonary host defense against *K. pneumoniae* infection, both male and female F1 mice were administered 1 *×* 10^3^ CFU of *K. pneumoniae* via oropharyngeal aspiration and sacrificed 48 h post infection. Pulmonary and splenic bacterial burden and lung damage of F1 human-microbiota mice were then assessed. F1 human alcohol-associated microbiota mice exhibited a significantly higher bacterial burden in both the lung (Figure 6A–D) and spleen (Figure 6E–H) when compared to F1 control microbiota mice. Precisely, human alcohol-associated microbiota mice had a significantly increased pulmonary bacterial burden in the combined group (Figure 6A), male mice (Figure 6B), and female mice (Figure 6C). In addition, there were clear effects on pulmonary bacterial load based on the original human donor (Figure 6D). Likewise, human alcohol-associated microbiota mice had significantly increased bacterial dissemination to distal organs (spleen) in the combined group (Figure 6E), although both male mice (Figure 6F) and female mice (Figure 6G) exhibited a nonsignificant trend towards increased bacterial burden. Consistent with previous data, alcohol-microbiota associated effects on bacterial dissemination were more pronounced in female mice. There were also clear effects on splenic bacterial load based on the original human donor (Figure 6H), suggesting a significant increase in pulmonary permeability following *K. pneumoniae* infection. To further determine whether increased bacterial dissemination was associated with lung-tissue damage, histological staining of lung tissue was performed. F1 human alcohol-associated microbiota mice exhibited a significant increase in lung injury post *K. pneumoniae* infection (Figure 7).

### 3.4. Human Alcohol-Associated Microbiota Mice Exhibit Increased Lung Leukocyte Recruitment and Inflammation

To define the effects of human alcohol-associated intestinal dysbiosis on pulmonary host defense against *K. pneumoniae* infection, male and female F1 human alcohol-associated microbiota mice were administered 1 × 10^3^ CFU of *K. pneumoniae* in the lung via oropharyngeal aspiration and sacrificed 48 h post infection. Analysis of BAL fluid from the lungs demonstrated that *K. pneumoniae* infection resulted in a significant increase in BAL protein, a marker of lung injury and leak (Figure 8). Human alcohol-associated microbiota mice had a significant increase in BAL protein levels in the combined group (Figure 8A). However, both male mice (Figure 8B) and female mice (Figure 8C) exhibited a nonsignificant trend in increased BAL protein. The original human donor sample also influenced BAL protein levels (Figure 8D).

F1 human alcohol-associated-microbiota mice also had decreased total numbers of leukocytes in their airways compared to control-microbiota mice (Figure 9A), with a non-significant decrease in total BAL cells in male mice (Figure 9B) but a significant decrease in total BAL cells in F1 human alcohol-associated microbiota female mice (Figure 9C). The original human donor sample (Figure 9D) also influenced total cell numbers. This corresponded with a nonsignificant decrease in BAL macrophages (Figure 9E) in both male (Figure 9F) and female (Figure 9G) mice and was not influenced by the original donor sample (Figure 9H). Conversely, F1 human alcohol-associated-microbiota mice also exhibited decreased numbers of BAL neutrophils (Figure 9I) with an even greater decrease of BAL neutrophils in males when compared to females (Figure 9J–K). BAL neutrophils were also not significantly influenced by the original donor sample (Figure 9L). There were no significant changes in lymphocytes across all groups.

Surprisingly, the levels of BAL IL-6, TNF-α, CXCL1, and IL-1β did not differ between alcohol-associated microbiota and control-microbiota mice when the data was combined for both genders in both groups (Figure 10A–D). However, F1 human alcohol-associated-microbiota mice exhibited a significant increase in the level of BAL GM-CSF (Figure 10E) with an even greater increase in the level of BAL GM-CSF in males when compared to females (Figure 9F–G). BAL GM-CSF levels were also influenced by the original donor sample (Figure 10H).

### 3.5. Human Alcohol-Associated Microbiota Mice have Increased Weight Loss following S. pneumoniae infection

*Streptococcus pneumoniae* is a Gram-positive organism and the leading etiological cause of pneumonia in alcohol-using individuals. As with the *K. pneumoniae* infection model, we sought to understand the role of human alcohol-associated dysbiosis on *S. pneumoniae* infection in both male and female F1 mice. Similar to previous studies, F1 human alcohol-associated mice were administered 4 × 10^8^ CFU of *S. pneumoniae* via oropharyngeal aspiration and weight change was determined 48 h post infection. Mice exhibited weight loss within the first 48 h post infection (Figure 11). Human alcohol-associated microbiota mice had significantly increased weight loss in the combined group (Figure 11A) and there was no difference between control and alcohol-associated male mice (Figure 11B). However, a nonsignificant weight loss in F1 human alcohol-associated microbiota female mice (Figure 11C) was observed, suggesting a potential gender effect following *S. pneumoniae* infection. In addition, there were clear donor-specific effects on weight loss post infection with *S. pneumoniae* (Figure 11D). Similar to the *K. pneumoniae* studies, weight loss data were consistent with the compromised physical appearance and decreased activity levels with F1 human alcohol-associated microbiota, with female mice having the most significant clinical signs. No difference in mortality was observed in the human alcohol-associated microbiota mice 48 h post infection.

### 3.6. Human Alcohol-Associated Microbiota Mice have Altered Lung Tissue Integrity and Increased S. pneumoniae burden

To evaluate the effects of human alcohol-associated dysbiosis on pulmonary host defense against *S. pneumoniae* infection, both male and female F1 mice were administered 4 × 10^8^ CFU of *S. pneumoniae* via oropharyngeal aspiration and sacrificed 48 h post infection. Pulmonary and splenic bacterial burden and lung damage of F1 human-microbiota mice were then assessed. F1 human alcohol-associated microbiota mice exhibited a significantly higher bacterial burden in both the lung (Figure 12A–D) and spleen (Figure 12E–H) when compared to F1 control microbiota mice. Precisely, human alcohol-associated microbiota mice had a significantly increased pulmonary bacterial burden in the combined group (Figure 12A), male mice (Figure 12B), and female mice (Figure 12C). In addition, there were distinct effects on pulmonary bacterial load due to the original human donor (Figure 12D). Likewise, human alcohol-associated microbiota mice had significantly increased bacterial dissemination to distal organs (spleen) in the combined group (Figure 12E) and in both male mice (Figure 12F) and female mice (Figure 12G). There was also a small effect on splenic bacterial load based on the original human donor (Figure 12H), which suggests that pulmonary permeability is increased following *S. pneumoniae* infection. To further determine whether increased bacterial dissemination was associated with lung tissue damage, histological staining of lung tissue was performed. F1 human alcohol-associated microbiota mice exhibited a significant increase in lung injury post-*Streptococcal* infection (Figure 13).

### 3.7. Human alcohol-associated Microbiota Mice have Increased Lung Leukocyte Recruitment and Inflammation

To define the effects of human alcohol-associated intestinal dysbiosis on the pulmonary immune response against *S. pneumoniae* infection, female F1 human alcohol-associated microbiota mice were administered 4 × 10^8^ CFU of *S. pneumoniae* in the lung via oropharyngeal aspiration and euthanized 48 h later. Analysis of BAL fluid from the lungs demonstrated that *S. pneumoniae* infection resulted in a significant increase in BAL protein, a marker of lung injury and leak (Figure 14A,B). F1 female human alcohol-associated microbiota mice also exhibited an increase in the total numbers of leukocytes in their airways compared to control-microbiota mice (Figure 14C,D), which corresponded with significant increases in BAL macrophages (Figure 14E,F) and BAL neutrophil numbers (Figure 14G,H). There were no significant changes in lymphocytes across the different groups. In addition to increased BAL cell numbers, F1 female human alcohol-associated microbiota mice had increased levels of IL-6 (Figure 15A,B), TNF-α (Figure 15C,D), and CXCL1 (Figure 15E,F) and a nonsignificant increase in IL-1β (Figure 15G,H) compared to control-microbiota mice.

### 3.8. Human Alcohol-Associated Microbiota Mice Exhibit Increased Drinking Preference and Anxiety-Like Behavior

Finally, we sought to determine whether the effects of human alcohol-associated intestinal dysbiosis extend beyond the gut–lung axis, namely to the gut–brain axis. To that end, we evaluated drinking preference and anxiety-like behavior in human alcohol-associated microbiota mice. Both male and female F1 mice were subjected to a 30-min marble-burying assessment prior to and following a 15-day 2-bottle choice alcohol-drinking paradigm. Interestingly, human alcohol-associated microbiota mice had a significant increase in baseline alcohol preference compared to control microbiota mice; however, no differences in drinking preference were observed at the end of the 15-day assessment (Figure 16A). Similarly, human alcohol-associated microbiota mice had a significant increase in baseline anxiety-like behavior when compared to control microbiota mice, but no differences in anxiety-like behavior were seen following 15 days of 2-bottle choice alcohol administration (Figure 16B). These data suggest that an alcohol-associated microbiota may be an important factor contributing to both anxiety-like behavior and drinking preference, both of which are commonly seen in individuals who consume alcohol chronically.

## 4. Discussion

It is now widely accepted that the intestinal microbiota plays a critical role in the host’s immune response to both bacterial and viral respiratory infections [2,27,28,29,30,31,32,33]. Regulation of the gut–lung axis has been led by work primarily using two model systems that include germ-free and antibiotic-treated mice. Both germ-free mice and antibiotic-treated mice are highly susceptible to pulmonary infection with leading bacterial pathogens, including *K. pneumoniae* and *S. pneumoniae* [27,28]. Further, fecal transplant of a healthy microbial community into these models bolsters pulmonary host defense, which suggests that optimal pulmonary host defense requires a “healthy” or diverse intestinal microbial community [27,28].

Alcohol increases the risk of pneumonia via impairment of several critical host defense mechanisms. First, chronic alcohol abuse significantly impairs mucus-facilitated clearance of bacterial pathogens from the upper airway [34,35]. Additionally, alcohol abuse increases the risk of aspiration of microbes from the upper alimentary tract. Taken together, these findings suggest that alcohol greatly increases the risk of pathogenic bacteria entering the lungs, thus increasing the risk of infection. Chronic alcohol consumption also suppresses tissue recruitment of neutrophils during infection and inflammation [36], as well as the phagocytic capacity of alveolar macrophages [37,38,39,40,41,42,43]. Chronic alcohol ingestion also decreases the differentiation and effector function of dendritic cells [44,45,46]. Alcohol use is also known to decrease the number of circulating lymphocytes and dysregulate Th1, Th2, and Th17 immune responses [40,43]. Because the impacts of alcohol on lung host defense are multi-factorial, and the gut–lung axis is now viewed as an important regulator of immune homeostasis, it is critical to define the role of alcohol-mediated gut dysbiosis on immune cell responses and host defense against bacterial infections that result from alcohol misuse. In addition, as access to alcohol treatment centers is often limited for many individuals, strategies that reduce the risk for alcohol-associated pneumonia are critically needed.

We have previously shown that alcohol-associated dysbiosis, independent of the direct effects of ethanol consumption, increased susceptibility to *K. pneumoniae* [2]. Alcohol-associated dysbiosis was also associated with impaired pulmonary immune cell recruitment, as well as marked intestinal permeability and inflammation [2]. Alcohol-associated dysbiosis also affects the number of mucosal associated invariant T-cells (MAIT cells) in mucosal tissues [47]. Furthermore, fecal microbiota transfer from alcohol-fed mice into alcohol-naïve mice resulted in a MAIT cell profile like those seen in alcohol-fed animals [47]. The differences in MAIT cells between alcohol- and control-fed mice were also mitigated by antibiotic treatment. Finally, we demonstrated that the intestinal microbiota can be targeted therapeutically to reduce host susceptibility to alcohol-associated bacterial pneumonia. More precisely, treatment of alcohol-fed mice with indole (a microbial metabolite) or with a probiotic cocktail improved pulmonary host defense by increasing pathogen clearance and pulmonary immune cell trafficking. Protective effects were, in part, mediated by AhR, as inhibition of AhR diminished the protective effects of indole [22]. Taken together, these studies demonstrate that (1) alcohol-associated dysbiosis increases bacterial pneumonia, (2) alcohol-associated microbial products increase systemic immune activation, and (3) microbial targeted therapies can mitigate the risk of alcohol-associated bacterial pneumonia. Importantly, all these studies are unified by the fact that alcohol-associated dysbiosis leads to immune cell dysfunction and impaired pulmonary recruitment [2,22,48,49].

Although preclinical studies have highlighted the importance of the intestinal microbiota in alcohol-related disorders, including behavioral consequences and immunological effects, these studies are limited in their ability to assess the direct effects of alcohol-associated dysbiosis independent of the direct effects of alcohol [1,2,3,5,6,22,47,48,49,50,51]. To date, these studies rely on associations, manipulating the microbiota with antibiotics, or conventionalization of germ-free mice [2,3,10]. While antibiotic cleansing and conventionalization of germ-free mice are valid models, they pose several challenges when trying to assess the immunological consequences of changes in microbiota community structure. For example, antibiotic treatment is known to have off-target effects and to influence a variety of immune cell populations [8]. Further, germ-free mice have aberrant immune development, and, upon conventionalization, exhibit marked immune reconstitution and activation for a period of time post conventionalization [52,53]. As such, there are currently no animal models that allow for the systematic evaluation of the effects of chronic alcohol abuse mediated by alcohol-associated dysbiosis independent of the direct effects of alcohol, without potential confounding effects of the treatment or mouse background on the immune system. To address this challenge, we developed a translationally relevant human alcohol-microbiota mouse model by colonizing germ-free mice with human fecal microbiota from individuals with high and low AUDIT scores and bred to produce human alcohol-associated microbiota or human control-microbiota F1 progenies. This model provides several advantages over our previous published work because: (1) we can evaluate the effects of human-specific microbial communities on disease progression and susceptibility; (2) both sexes of mice can be evaluated; (3) confounding from repeated conventionalization or by off target effects due to antibiotics are eliminated; (4) individual donor (human-level characteristics) can be evaluated alongside group/disease effects; (5) microbiota traits are inherited across generations of mice; and (6) both behavioral and physiological traits can be evaluated.

## 5. Conclusions

Disease-associated microbial communities impair host immune responses, independent of disease-associated sequelae. Here, we established that human alcohol-associated microbiota mice have increased susceptibility to both *K. pneumoniae* and *S. pneumoniae* pneumonia when compared to mice colonized with a human control “healthy” microbiota. Human alcohol-associated microbiota mice have increased mortality, pulmonary bacterial burden, bacterial dissemination, lung damage/leak, and increases in pulmonary immune cell infiltration and inflammation. However, many of the mechanisms by which the microbiota controls or alters host defense against respiratory infections are still ill-defined. Determining and characterizing the mechanisms that facilitate crosstalk between innate lymphocytes, the microbiota, and indole in a highly translatable model of human alcohol-associated dysbiosis is our primary future goal. It is likewise important to understand the opposing effects between *K. pneumoniae* and *S. pneumoniae* in the BAL analysis of cell infiltrates and chemokines, as well as to evaluate the pulmonary cellular infiltration and inflammation in male mice infected with *S. pneumoniae*. In turn, we envision that these efforts will lead to new screening methods to identify at-risk populations and subsequently to innovative strategies to reduce the burden of CAP.

## Figures and Tables

**Figure 1 cells-12-02267-f001:**
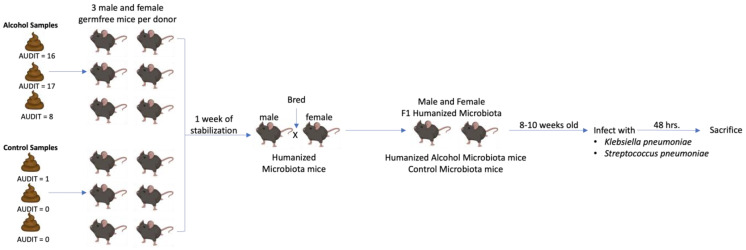
Experimental Schema.

**Figure 2 cells-12-02267-f002:**
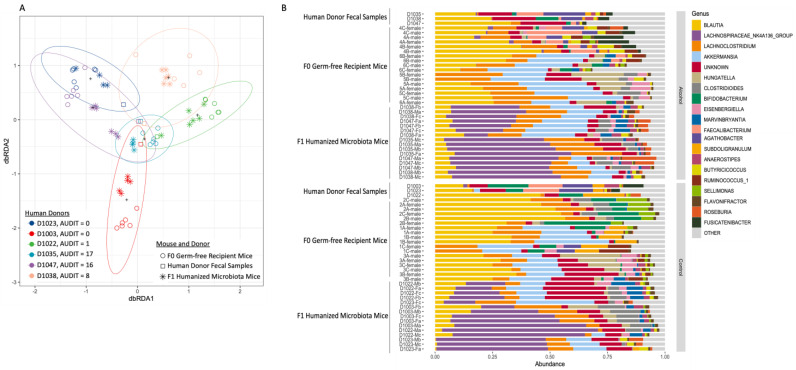
Microbial community structure is maintained in the F1 generation of human alcohol-associated microbiota mice, as shown by the 16s sequencing of human donor fecal samples and cecal microbial community from F0 and F1 conventionalized germ-free mice. (**A**) Beta diversity of human, F0, and F1 human-microbiota associated mice, as determined by distance-based redundancy analysis (dbRDA) on sample-wise Bray–Curtis dissimilarity distances. (**B**) Relative abundance of microbial taxa in human fecal samples, as well as cecal samples from F0, and F1 human-microbiota associated mice. *n* = 5–6/group.

**Figure 3 cells-12-02267-f003:**
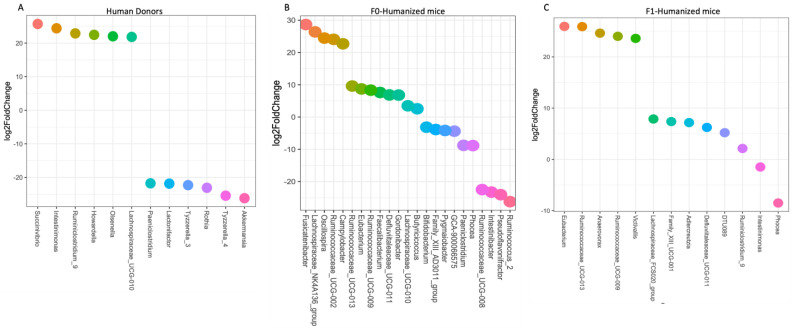
Differentially abundant microbial taxa structure in human alcohol-associated microbiota mice. Differentially abundant ASVs as determined by DESeq2. (**A**) Human donor fecal samples, (**B**) F0 conventionalized mice, and (**C**) F1 human-microbiota associated mice. *n* = 5–6/group.

**Figure 4 cells-12-02267-f004:**
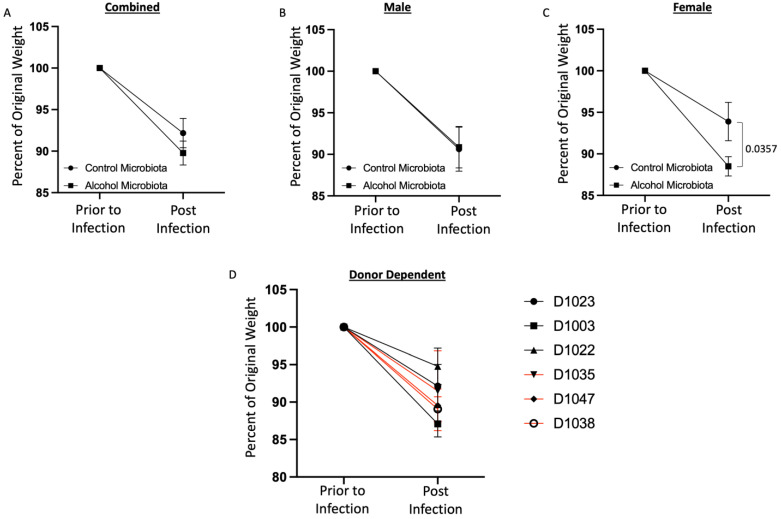
Human alcohol-associated microbiota mice have increased weight loss following *K. pneumoniae* infection. F1 microbiota mice were infected with *K. pneumoniae*, and weight loss was assessed daily for 48 h post infection. Post-infection weight loss in (**A**) combined male and female mice, (**B**) male mice, and (**C**) female mice. (**D**) Donor-dependent weight change in combined male and female mice post infection. Dots represent the mean and SEM per group (*n* = 20 control-microbiota F1 mice, and *n* = 20 alcohol-associated F1 mice). Red lines indicate F1 alcohol-associated microbiota mice. *p* values are indicated in the figure and were determined by two-way ANOVA with a post hoc Sidak’s multiple comparisons correction.

**Figure 5 cells-12-02267-f005:**
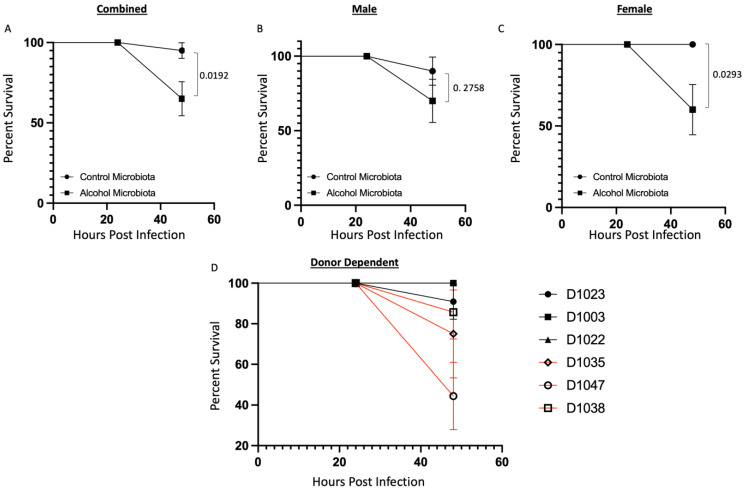
Human alcohol-associated microbiota mice have increased mortality following *K. pneumoniae* infection. F1 microbiota mice were infected with *K. pneumoniae*, and survival was assessed 48 h post infection. Survival post infection in (**A**) combined male and female mice, (**B**) male mice, and (**C**) female mice. (**D**) Donor-dependent survival in combined male and female mice post infection. Dots represent the mean and SEM per group (*n* = 20 control-microbiota F1 mice, and *n* = 20 alcohol-associated F1 mice). Red lines indicate F1 alcohol-associated microbiota mice. *p* values are indicated in the figure and were determined by the Log-rank (Mantel–Cox) test.

**Figure 6 cells-12-02267-f006:**
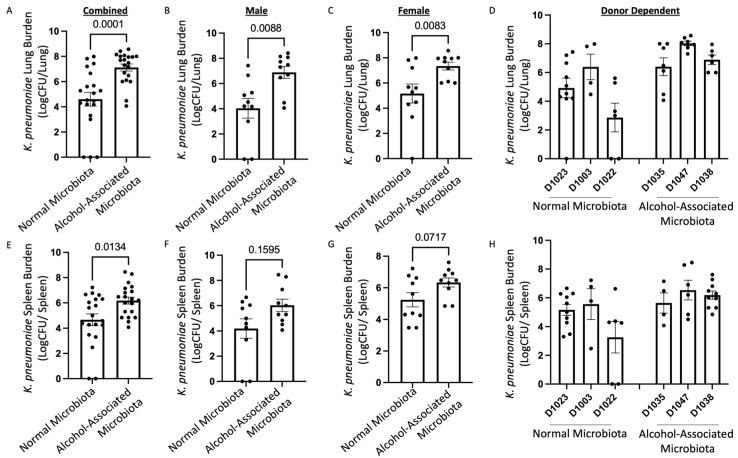
Human alcohol-associated microbiota mice have increased pulmonary bacterial burden and dissemination. F1 Microbiota mice were infected with *K. pneumoniae* and bacterial burden was assessed. Log transformation burden of *K. pneumoniae* in the lungs of (**A**) combined male and female F1 mice, (**B**) male mice, (**C**) female mice, and (**D**) donor-dependent pulmonary burden. Log transformation burden of *K. pneumoniae* in the spleens of (**E**) combined male and female F1 mice, (**F**) male mice, (**G**) female mice, and (**H**) donor-dependent pulmonary burden. Bars represent the mean ± SEM and dots represent individual mice. *p* values are indicated in the figure and were determined by Welch’s t-test.

**Figure 7 cells-12-02267-f007:**
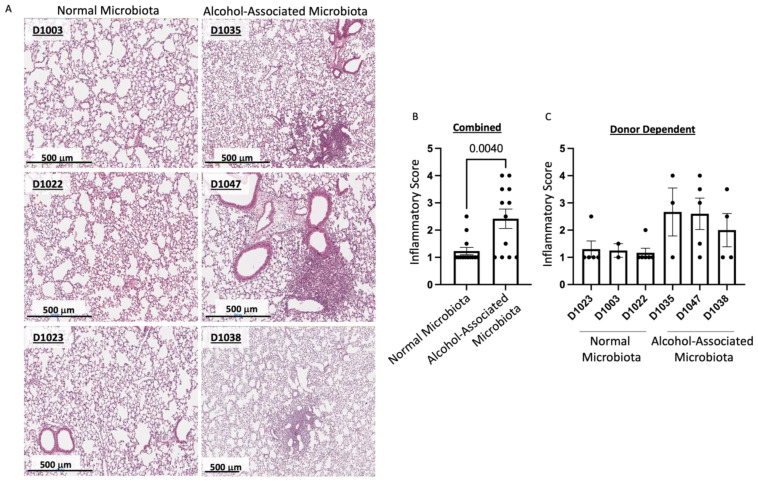
Human alcohol-associated microbiota mice have increased pulmonary damage. F1 Microbiota mice were infected with *K. pneumoniae*, and pulmonary damage was assessed. (**A**) Representative lung H&E images at 20× magnification. Lung inflammatory scores via H&E histology of (**B**) combined male and female F1 mice and (**C**) donor-dependent pulmonary damage. Bars represent the mean ± SEM, and dots represent individual mice. *p* values are indicated in the figure and were determined by Welch’s t-test.

**Figure 8 cells-12-02267-f008:**
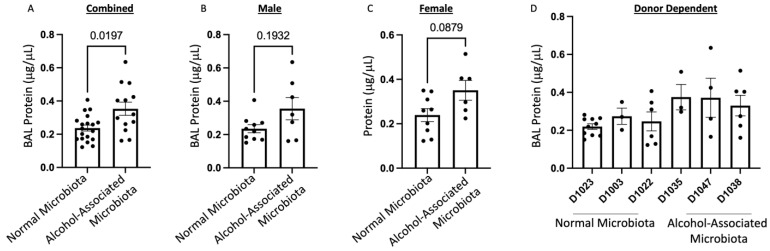
Human alcohol-associated microbiota mice have decreased barrier function. F1 microbiota mice were infected with *K. pneumoniae*, and the level of BAL protein was determined in the lungs of (**A**) combined male and female F1 mice, (**B**) male mice, (**C**) female mice, and (**D**) donor-dependent pulmonary BAL protein levels. Bars represent the mean ± SEM and dots represent individual mice. *p* values are indicated in the figure and were determined by Welch’s t-test.

**Figure 9 cells-12-02267-f009:**
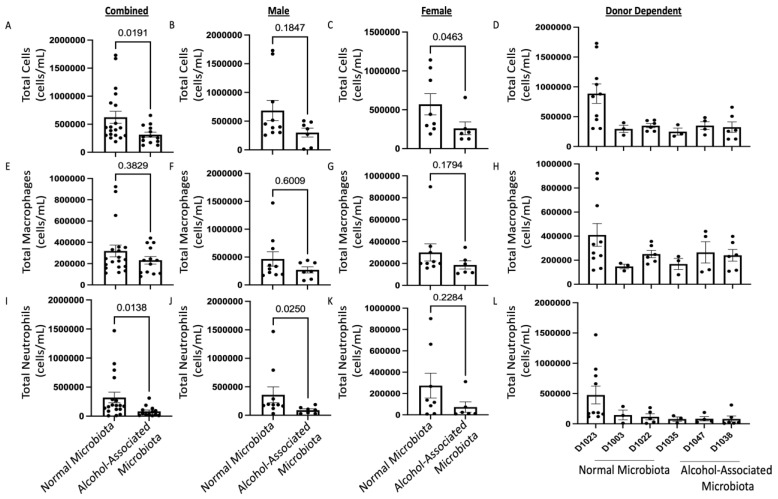
Human alcohol-associated microbiota mice have decreased pulmonary immune cell numbers. F1 microbiota mice were infected with *K. pneumoniae* and the numbers of BAL immune cells were assessed. Total BAL counts were determined in the lungs of (**A**) combined male and female F1 mice, (**B**) male mice, (**C**) female mice, and (**D**) donor dependent. Total macrophages post infection were determined in the lungs of (**E**) combined male and female F1 mice, (**F**) male mice, (**G**) female mice, and (**H**) donor dependent (**D**). Total neutrophils post infection were determined in the lungs of (**I**) combined male and female F1 mice, (**J**) male mice, (**K**) female mice, and (**L**) donor dependent. Bars represent the mean ± SEM, and dots represent individual mice. *p* values are indicated in the figure and were determined by Welch’s t-test.

**Figure 10 cells-12-02267-f010:**
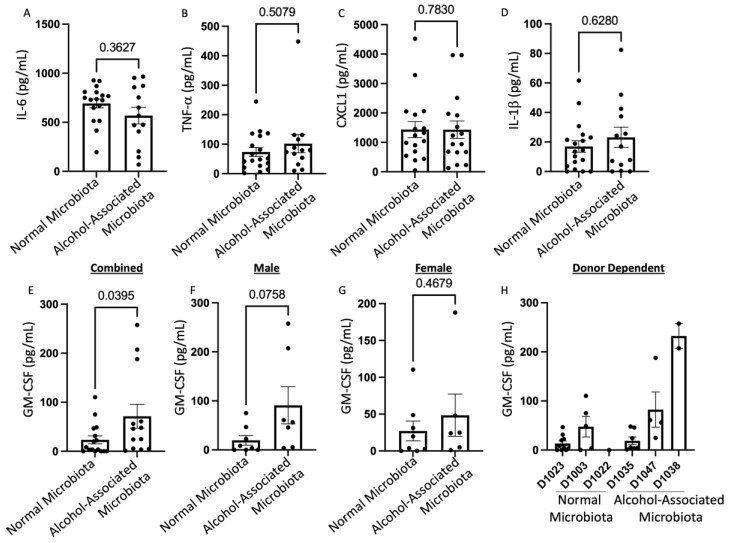
Human alcohol-associated microbiota mice have altered pulmonary inflammation. F1 microbiota mice were infected with *K. pneumoniae*, and the levels of BAL cytokines/chemokines were assessed. Total BAL (**A**) IL-6, (**B**) TNF-α, (**C**) CXCL1, and (**D**) IL-1β were determined in the lungs of combined male and female F1 mice. Total BAL GM-CSF levels were determined in the lungs of (**E**) combined male and female F1 mice, (**F**) male mice, (**G**) female mice, and (**H**) donor dependent. Bars represent the mean ± SEM, and dots represent individual mice. *p* values are indicated in the figure and were determined by Welch’s t-test.

**Figure 11 cells-12-02267-f011:**
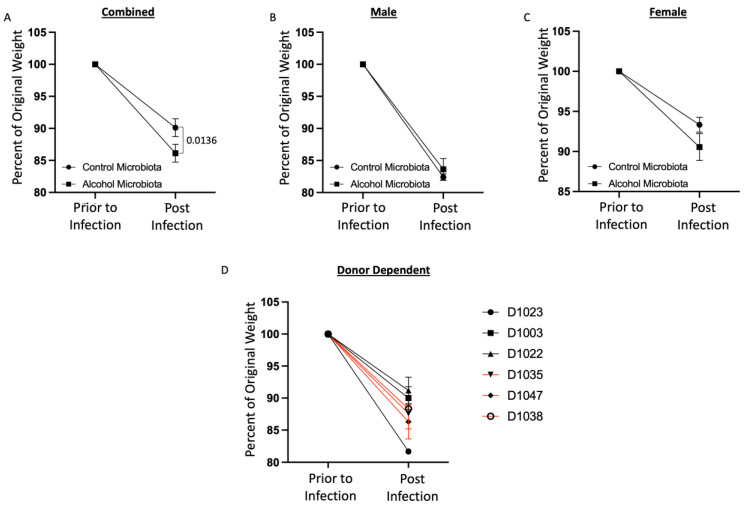
Human alcohol-associated microbiota mice exhibit increased weight loss following *S. pneumoniae* infection. F1 microbiota mice were infected with *S. pneumoniae* and weight loss was assessed out to 48 h post infection. Post-infection weight loss in (**A**) combined male and female mice, (**B**) male mice, and (**C**) female mice. (**D**) Donor-dependent weight change in combined male and female mice post infection. Dots represent the mean and SEM per group (*n* = 18 control-microbiota F1 mice and *n* = 25 alcohol-associated F1 mice). Red lines indicate F1 alcohol-associated microbiota mice. *p* values are indicated in the figure and were determined by two-way ANOVA with a post-hoc Sidak’s multiple comparisons correction.

**Figure 12 cells-12-02267-f012:**
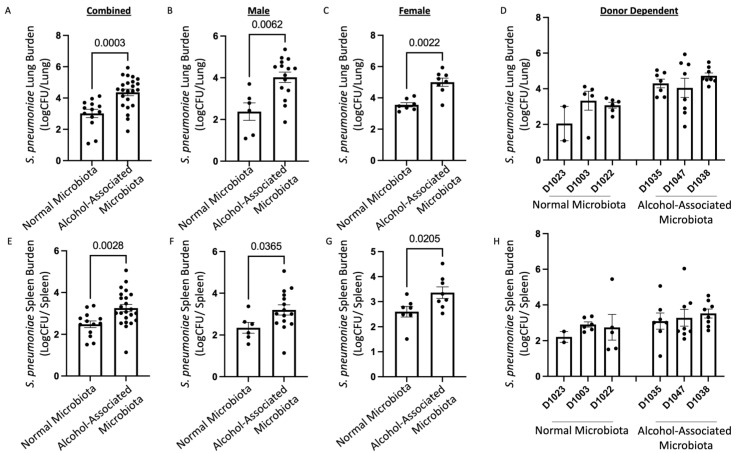
Human alcohol-associated microbiota mice have increased pulmonary bacterial burden and dissemination. F1 microbiota mice were infected with *S. pneumoniae* and bacterial burden was assessed. Log transformation burden of *S. pneumoniae* in the lungs of (**A**) combined male and female F1 mice, (**B**) male mice, (**C**) female mice, and (**D**) donor-dependent pulmonary burden. Log transformation burden of *S. pneumoniae* in the spleens of (**E**) combined male and female F1 mice, (**F**) male mice, (**G**) female mice, and (**H**) donor-dependent pulmonary burden. Bars represent the mean ± SEM, and dots represent individual mice. *p* values are indicated in the figure and were determined by Welch’s t-test.

**Figure 13 cells-12-02267-f013:**
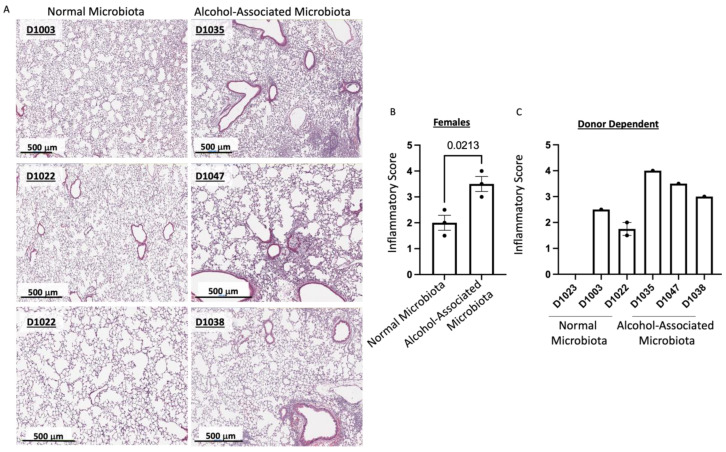
Human alcohol-associated microbiota mice have increased pulmonary damage. F1 microbiota mice were infected with *S. pneumoniae*, and pulmonary damage was assessed. (**A**) Representative lung H&E images at 20× magnification. Lung inflammatory scores of (**B**) female F1 mice, and (**C**) donor-dependent pulmonary damage. Bars represent the mean ± SEM, and dots represent individual mice. *p* values are indicated in the figure and were determined by Welch’s t-test.

**Figure 14 cells-12-02267-f014:**
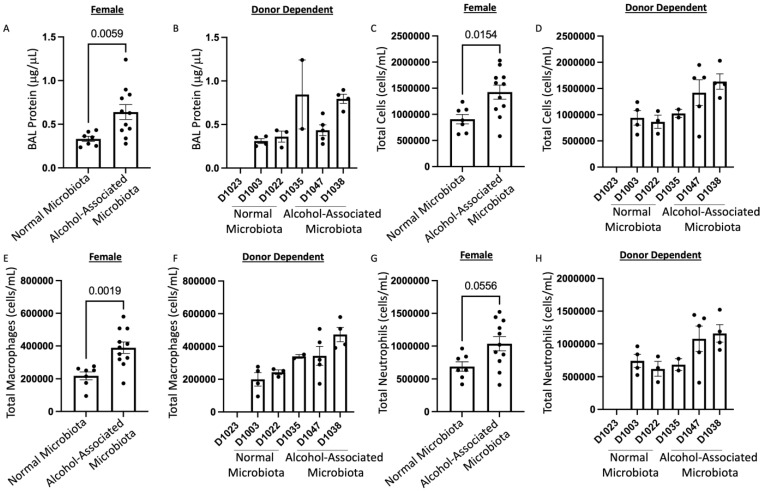
Human alcohol-associated microbiota mice have increased pulmonary leak and immune cell numbers. F1 microbiota mice were infected with *S. pneumoniae* and the level of BAL protein was determined in the lungs of (**A**) female F1 mice, and (**B**) donor-dependent pulmonary BAL protein levels. The numbers of BAL immune cells were assessed. Total BAL counts were determined in the lungs of (**C**) female F1 mice and (**D**) donor-dependent mice. Post-infection macrophages were enumerated in the lungs of (**E**) female F1 mice, and (**F**) donor-dependent mice. Post-infection neutrophils were also enumerated in the lungs of (**G**) female F1 mice, and (**H**) donor dependent mice. Bars represent the mean ± SEM, and dots represent individual mice. *p* values are indicated in the figure and were determined by Welch’s t-test.

**Figure 15 cells-12-02267-f015:**
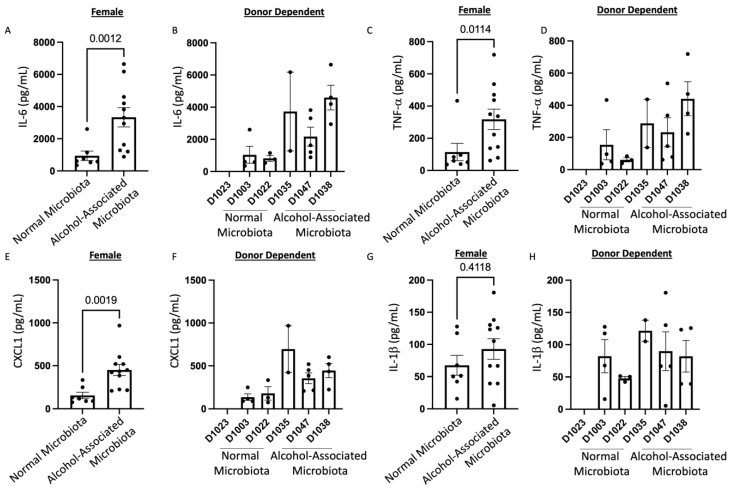
Human alcohol-associated microbiota mice have increased pulmonary inflammation. F1 microbiota mice were infected with *S. pneumoniae* and the levels of BAL cytokines/chemokines were assessed. Total BAL (**A**,**B**) IL-6, (**C**,**D**) TNF-α, (**E**,**F**) CXCL1, and (**G**,**H**) IL-1β were determined in the lungs of female F1 mice. Bars represent the mean ± SEM and dots represent individual mice. *p* values are indicated in the figure and were determined by Welch’s t-test.

**Figure 16 cells-12-02267-f016:**
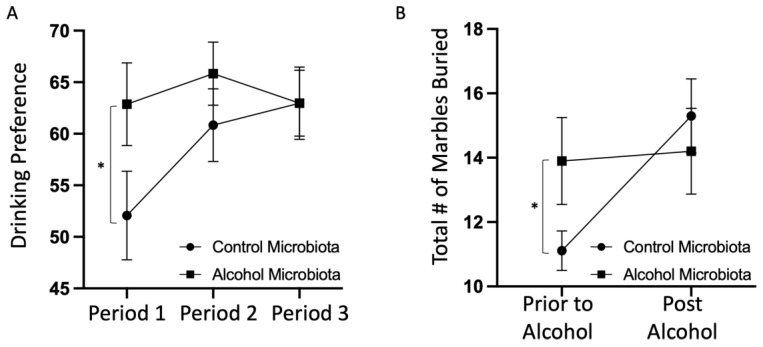
Human alcohol-associated microbiota mice have increased drinking preference and anxiety-like behavior. Male and female F1 alcohol-associated microbiota mice were subjected to a 30-min marble-burying assessment prior to and following a 15-day 2-bottle choice alcohol-drinking paradigm. (**A**) 2-bottle choice alcohol-drinking preference, and (**B**) marble-burying score. Dots represent the mean and SEM per group (*n* = 10 (5 male and 5 female) control-microbiota F1 mice, and *n* = 10 (5 male and 5 female) alcohol-associated F1 mice). *p* values are indicated with a *, corresponding to *p* = 0.03 and *p* = 0.05 for drinking preference and marble-burying, respectively. *p* values were determined by the Mann–Whitney test.

**Table 1 cells-12-02267-t001:** Demographics of human fecal samples.

Sample ID	Gender	Age	AUDIT C	AUDIT Total
D1022	Female	60	1	1
D1003	Female	35	0	0
D1023	Male	63	0	0
D1038	Male	63	2	8
D1035	Male	63	9	17
D1047	Male	46	6	16

## Data Availability

Sequencing data is deposited in the National Center for Biotechnology Information Sequence Read Archive (BioProject ID: PRJNA997320). All additional data supporting the findings of this study are available within the paper and are available upon request.

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
