# Peer review of "Human Alcohol-Microbiota Mice have Increased Susceptibility to Bacterial Pneumonia"

_cells, 2023, doi:10.3390/cells12182267_

Round 1

Reviewer 1 Report

The authors have done a great job of using germ-free mice to understand the role of alcohol-associated dysbiosis. The methods and the results clearly show the role of alcohol-associated microbiome population in causing inflammation and other immune responses. As mentioned in the conclusion, the mechanism of the same needs to be elucidated, and is the logical next step in the study.

Author Response

 We thank the reviewer for their review of the manuscript and their feedback.

Reviewer 2 Report

This is a very well designed and written manuscript.

One minor comment:

Please use consistent bacterial names throughout the manuscript: Klebsiella pneumoniae (K. pneumoniae) and Streptococcus pneumoniae (S. pneumoniae). Avoid single designations Klebsiella and Streptococcus.

(e.g. lines: 29, 33, 34, 60, 138, 239, 246, 247, 254, 255, 278, 297, 316, 378, 379, 390, 391, 405, 407, 443, 575, 582, 584)

Author Response

We thank the reviewer for their review of the manuscript and their feedback. We have amended all instances of the inconsistent bacterial names. K. pneumoniae and S. pneumoniae are used throughout the manuscript. 

Reviewer 3 Report

The intriguing manuscript titled "Human alcohol-microbiota mice have increased susceptibility to bacterial pneumonia" establishes a humanized alcohol-microbiota mouse model to systematically assess the immunological impacts of chronic alcohol abuse mediated by intestinal dysbiosis. I think this study is innovative and novel, several conclusions lack robust scientific support. If the conclusion of the article pertains to model establishment, the existing research findings are insufficient to meet the evaluation requirements for the new model's establishment, necessitating a comprehensive evaluation of the model from various aspects. Therefore, I believe that after a major revision, the manuscript could be considered for acceptance or rejection.

 1. Please provide further clarification on the necessity and gaps filled by establishing humanized alcohol-microbiota mouse model.

2. The statement regarding F1 offspring colonized with fecal microbiota from individuals with high AUDIT scores showing increased susceptibility to Klebsiella and Streptococcus pneumonia requires support from experiments involving mono-culture or dual-culture colonization. Without this, the conclusion may lack rigor.

3. Kindly elaborate on the aspects through which “susceptibility” is manifested, requiring further explanation within the context.

4. Intestinal microbiota is often greatly influenced by the environment; therefore, human fecal samples should offer more information, such as dietary intake, medication history, and other conditions.

5. Conducting 16s analysis on microbiota before and after colonization would help determine colonization rates.

6. Enhance the description of the experimental process and the analysis of microbiota. Figure 3 suggests continued differences in microbial communities before and after model construction. Does this imply unsuccessful human fecal colonization?

7. Could the colonization of these two microorganisms trigger other physiological changes? This aspect should be addressed in the discussion section.

The quality of English language in the manuscript is generally good. 

Author Response

We thank the reviewer for their review of the manuscript and their feedback. We have addressed all of the reviewer's comments below.

Comment 1: Please provide further clarification on the necessity and gaps filled by establishing humanized alcohol-microbiota mouse model.

Response: We have added a sentence to the introduction to highlight the gap in knowledge regarding human alcohol-associated dysbiosis. See lines 38-49.

“Numerous preclinical studies have highlighted the importance of the intestinal microbiota in alcohol-related disorders. Alcohol-induced changes in the gastrointestinal microbiota composition and metabolic function are associated with 1) behavior consequences, 2) physiological alterations, and 3) immunological effects [1-7]. However, these studies are limited in their ability to assess the direct effects of alcohol-associated dysbiosis independent of the direct effects of alcohol, as they rely on either association studies, or manipulating the microbiota with antibiotics, which is known to have off-target effects [8]. As such, there are currently no animal models that allow for the systematic evaluation of the effects of chronic alcohol abuse that are mediated by alcohol-associated dysbiosis independent of the direct effects of alcohol. This is especially true for evaluating the effects of human alcohol-associated dysbiosis, as only a handful of studies have sought to understand the functional consequences of human alcohol-associated dysbiosis [1,5,9,10].”

Comment 2: The statement regarding F1 offspring colonized with fecal microbiota from individuals with high AUDIT scores showing increased susceptibility to Klebsiella and Streptococcus pneumonia requires support from experiments involving mono-culture or dual-culture colonization. Without this, the conclusion may lack rigor.

Response: Germ-free mice are colonized with human fecal microbiota from three different AUDIT high individuals, as well as from three different AUDIT low individuals, as such it’s unclear what an experimental setup would be used for dual or monoculture, in terms of colonization. Assuming monoculture and dual-culture referees to Klebsiella and Streptococcus, the mice are not colonized with these organisms, but infected with the organisms for 48 hrs. and sacrificed.

Comment 3: Kindly elaborate on the aspects through which “susceptibility” is manifested, requiring further explanation within the context.

Response: Mice colonized with microbiota from human AUDIT high individuals have increased susceptibility to both K. pneumoniae and S. pneumoniae, as determined by increased pulmonary and splenic bacterial burden, increased lung inflammation and leak, altered cellular infiltrates, and increased lung damage. This is shown in Figures 4-15.

Comment 4: Intestinal microbiota is often greatly influenced by the environment; therefore, human fecal samples should offer more information, such as dietary intake, medication history, and other conditions.

Response: Thank you for the comment, while the microbiota is influenced by diet and other environmental factors, we do not have access to many of these factors from the de-identified samples. However, the groups of germ-free mice were colonized with human fecal microbiota from three different AUDIT high individuals, as well as from three different AUDIT low individuals. This provides additional individual donor level rigor, as we still see group effects even when not accounting for diet, medication, or other co-morbid conditions associated with the different donors.

  1. Conducting 16S analysis on microbiota before and after colonization would help determine colonization rates.

Response: 16S analysis of the microbiota samples from before and after colonization is shown in Figure 2 and 3. Colonization rate, if referred to Klebsiella and Streptococcus, the mice are not colonized with these organisms.

  1. Enhance the description of the experimental process and the analysis of microbiota. Figure 3 suggests continued differences in microbial communities before and after model construction. Does this imply unsuccessful human fecal colonization?

Response: Yes, there are differences between the original human sample and the F0 and F1 generation of mice, however, the mice still cluster with respect to the original human fecal sample. No colonization method is going to produce a complete recapitulation of the original human sample, as many factors are associated with colonization.

  1. Could the colonization of these two microorganisms trigger other physiological changes? This aspect should be addressed in the discussion section.

Response: The F1 mice are not colonized with Klebsiella or Streptococcus. Figures 4-15 depict the host response to infection with these organisms in mice colonized with human GI microbiota samples.